# Neuroplastin Expression in Male Mice Is Essential for Fertility, Mating, and Adult Testosterone Levels

**DOI:** 10.3390/ijms25010177

**Published:** 2023-12-22

**Authors:** Juanjuan Chen, Xiao Lin, Soumee Bhattacharya, Caroline Wiesehöfer, Gunther Wennemuth, Karin Müller, Dirk Montag

**Affiliations:** 1Neurogenetics, Leibniz Institute for Neurobiology, Brenneckestr. 6, D-39118 Magdeburg, Germany; juanjuan.chen@lin-magdeburg.de (J.C.); linxiao@gdiist.cn (X.L.); soumee.bhattacharya@iconeus.com (S.B.); 2Department of Anatomy, University Hospital, University Duisburg-Essen, Hufelandstr. 55, D-45147 Essen, Germany; caroline.wiesehoefer@uk-essen.de (C.W.); gunther.wennemuth@uk-essen.de (G.W.); 3Leibniz Institute for Zoo and Wildlife Research IZW, Alfred-Kowalke-Str. 17, D-10315 Berlin, Germany; mueller@izw-berlin.de

**Keywords:** testosterone, infertility, testis, mounting behavior, neuroplastin, Leydig cells

## Abstract

Male reproduction depends on hormonally driven behaviors and numerous genes for testis development and spermatogenesis. Neuroplastin-deficient (*Nptn^−/−^*) male mice cannot sire offspring. By immunohistochemistry, we characterized neuroplastin expression in the testis. Breeding, mating behavior, hormonal regulation, testicular development, and spermatogenesis were analyzed in cell-type specific neuroplastin mutant mice. Leydig, Sertoli, peritubular myoid, and germ cells express Np, but spermatogenesis and sperm number are not affected in *Nptn^−/−^* males. Neuroplastin lack from CNS neurons or restricted to spermatogonia or Sertoli cells permitted reproduction. Normal luteinizing hormone (LH) and follicle-stimulating hormone (FSH) blood levels in *Nptn^−/−^* males support undisturbed hormonal regulation in the brain. However, *Nptn^−/−^* males lack mounting behavior accompanied by low testosterone blood levels. Testosterone rise from juvenile to adult blood levels is absent in *Nptn^−/−^* males. LH-receptor stimulation raising intracellular Ca^2+^ in Leydig cells triggers testosterone production. Reduced Plasma Membrane Ca^2+^ ATPase 1 (PMCA1) in *Nptn^−/−^* Leydig cells suggests that *Nptn^−/−^* Leydig cells produce sufficient testosterone for testis and sperm development, but a lack of PMCA-Np complexes prevents the increase from reaching adult blood levels. Behavioral immaturity with low testosterone blood levels underlies infertility of *Nptn^−/−^* males, revealing that Np is essential for reproduction.

## 1. Introduction

Infertility is a complex multifactorial condition and can result from both genetic and environmental causes. Genetic defects in humans, accounting for 15–30% of male infertility, majorly elicit malfunctioning of sex determination and/or development, sperm production and/or function, and a plethora of other physiological abnormalities, including endocrinopathies (for review, see [1,2,3]). Previous reports have identified several autosomal gene mutations and polymorphisms that affect male fertility, e.g., the cystic fibrosis transmembrane conductance regulator (*CFTR*) gene encoding a glycosylated transmembrane chloride channel affecting the vas deferens (for review, see [4]), the *SHBG* gene encoding the glycoprotein sex hormone-binding globulin (SHBG), or genes associated with spermatogenic failure (for review, see [5]). However, approximately 25% of the cases of clinical infertility are idiopathic, reflecting our relatively insufficient understanding of the molecular mechanisms underlying it (for review, see [1,3]).

Recently, we reported that the constitutive lack of neuroplastin (Np) in mice, in addition to learning and memory deficits, also affects the endocrine system [6]. We observed elevated corticosterone levels, decreased levels of corticotropin-releasing hormone mRNA, and behavioral symptoms associated with stress, anxiety, and/or depression, pathological conditions that are usually neuroendocrine in nature [6]. Furthermore, neuroplastin-deficient (*Nptn^−/−^*) male mice do not sire offspring, demonstrating a role of Np in reproduction [6].

Np is a phylogenetically conserved type I glycoprotein cell adhesion molecule belonging to the immunoglobulin (Ig) superfamily [7] (for a recent review, see [8]). The major isoforms Np55 and Np65 consist of two or three Ig-like extracellular domains, a single transmembrane domain, and a short cytoplasmic tail [7]. The shorter isoform Np55 is widely expressed in most organs, whereas the 3 Ig-isoform Np65 is expressed only in neurons. Polymorphisms in the regulatory region of the human *NPTN* gene correlated with cortical thickness and intellectual abilities in adolescents [9] and were detected in individuals suffering from schizophrenia [10]. Recently, the human *NPTN* gene has been associated with neuropsychiatric diseases (for review, see [11]), heart rate [12], lung cancer [13], and *Nptn* in mice with hearing deficits [11,14,15] and T-cell activation [16]. Np undergoes multiple interactions with protein binding partners (for review, see [8]). Importantly, Np promotes the expression of Plasma Membrane Calcium ATPases (PMCA) and forms functional complexes [6,17,18] for extrusion of intracellular Ca^2+^ related Np to termination and control of Ca^2+^-mediated signal transduction (for review, see [19]). Np is a close paralog of CD147 (basigin/EMMPRIN) [20]. However, basigin-deficient mice of both sexes are severely affected and do not reproduce [21,22], whereas *Nptn^−/−^* females are fully fertile [6].

In this study, we investigated the essential role of neuroplastin in male fertility. We analyzed Np expression in the mouse testis and studied *Nptn^−/−^* male mice using histological, hormonal, and behavioral analyses. The production of morphologically normal sperm and testis development in the absence of Np is contrasted by the complete absence of mating behavior of adult *Nptn^−/−^* males. We show that the testosterone levels in the blood of adult *Nptn^−/−^* males do not reach adult wild-type concentrations but remain at juvenile levels, being insufficient for expression of normal mating behavior. We suggest that LH-signaling triggering testosterone production by adult Leydig cells is compromised caused by insufficient Ca^2+^ extrusion via Np-associated PMCA1.

## 2. Results

### 2.1. Nptn^−/−^ Male Mice Do Not Produce Offspring

Confirming our previous observations, *Nptn^−/−^* male mice did not sire any offspring, nor was any pregnancy detected (see Materials and Methods). In contrast, when *Nptn^+/−^* males were paired with *Nptn^+/−^* females or with C57BL/6NCrl females, the size and frequency of litters produced were similar to C57BL/6NCrl breedings.

#### 2.1.1. *Nptn^−/−^* Males Display Abnormal Mating Behavior

Therefore, we investigated the mating behavior of *Nptn^−/−^* male mice in detail. On exposure to a socially naive female in their home cage, *Nptn^−/−^* males did not attempt to mount the females, resulting in a complete absence of mounts with thrusts compared to *Nptn^+/+^* littermate males, which all mounted within short latencies (12.4 ± 1.21 s latency to first mount; 4 ± 1.18 mounts/30 min). Additionally, the *Nptn^−/−^* male mice displayed aberrant investigatory behavior by spending significantly reduced time in anogenital sniffing (84 ± 24 s) as compared to the wild-type *Nptn^+/+^* (168 ± 12 s) counterparts (Figure 1A, one-way ANOVA F_(1,9)_ = 7.778, *p* = 0.0211). The total interaction time (=body sniffing + anogenital sniffing + direct contact) revealed no difference between the genotypes (Figure 1B, *Nptn^+/+^* 5.4 min ± 1.09; *Nptn^−/−^* 4.3 min ± 1.12).

#### 2.1.2. External Genitalia, Testis, and Sperm in *Nptn^−/−^* Male Mice

In 4–5-month-old adult *Nptn^−/−^* males, the appearance of the external genitalia, the weight, and the morphology of the testis were comparable to their wild-type *Nptn^+/+^* littermates (Figure 2). Additionally, the testis weight during male development and of adult *Nptn^−/−^* males and wild-type littermates was similar (Figure 2C). The testicular histology did not reveal significant differences with respect to the size and morphological appearance of cross sections observed between adult *Nptn^−/−^* and *Nptn^+/+^* males (Figure 2D). The average number of seminiferous tubules and their perimeter were quantified per transverse section obtained from the testis and found to be comparable between the genotypes (Figure 2F). Furthermore, the amount and concentration of sperm isolated from the cauda of the epididymis from adult *Nptn^−/−^* (140 days old, *n* = 6, 18.25 ± 0.78 × 10^6^/cauda) were similar to the sperm derived from wild-type littermate males (*n* = 6, 15.08 ± 1.30 × 10^6^/cauda). Isolated sperm from *Nptn^+/+^* and *Nptn^−/−^* male mice showed comparable beat frequencies, which were similarly increased after activation with 15 mM bicarbonate (Figure 3A). The spermatogenetic activity assessed by the ploidy status of testicular cell nuclei at 20, 28, 34, and 45 days of age revealed that the proportion of haploid (1c), diploid (2c), and tetraploid (4c) cells from *Nptn^+/+^* and *Nptn^−/−^* male mice did not differ (Figure 3B).

#### 2.1.3. Neuroplastin Is Expressed in Testis and Present on Sperm of Male Mice

By RT-PCR, we confirmed the expression of neuroplastin mRNA in the testis of adult male mice (Figure 3C), consistent with the expression of Np in rat testis [23] and single-cell RNA sequencing data by Green et al. [24], who reported *Nptn* mRNA expression by several cell types in the testis, e.g., 5x higher expression in peritubular myoid cells compared to spermatogonia, Leydig, and Sertoli cells and expressed also in spermatocytes and spermatids of male mice. At the protein level, polyclonal antibodies against Np reveal two bands at 55 and 65 kDa in brain and a single band of 50 kDa in testis and of 45 kDa in sperm from *Nptn^+/+^* male mice on Western blots (Figure 3D). The brain-specific Np65 isoform is detected by Np65-specific antibodies in the brain but not in testis tissue or in sperm. Cell type-specific glycosylation of the 28 kDa core protein, which is expressed in most tissues [7,23], leads to the observed band size around 50 kDa in testis and 45 kDa in sperm. On isolated wild-type sperm, Np was detectable as punctate surface staining on sperm heads and tails (Figure 3E) but could not be detected in sperm of *Nptn^−/−^* male mice.

Expression of Np in mouse testis was detected by immunohistochemistry in the seminiferous tubules of *Nptn^+/+^* males, in highest amounts in the outermost layer of the tubules, whereas Np could not be detected in the testis from *Nptn^−/−^* males (Figure 4 and Figure 5). In adult male *Nptn^+/+^* mice, Np expression was detected in myoid cells, Leydig cells, spermatogonia, and spermatocytes, as well as in Sertoli cells. Noteworthy, the expression patterns revealed by antibodies detecting alpha-smooth muscle actin (myoid cells), CYP11A (Cytochrome P450 Family 11 Subfamily A Member 1, Leydig cells) (Figure 4), Stra8 (spermatogonia and spermatocytes), and Sox9 (Sertoli cells) (Figure 5) were undistinguishable between *Nptn^+/+^* and *Nptn^−/−^*, indicating the normal development and maturation of the testis in *Nptn^−/−^* male mice.

### 2.2. Ablation of Nptn-Expression in Spermatogonia, Sertoli-Cells, or Central Nervous System

Directing expression of Cre recombinase by the Stra8 promoter in *Nptn^loxloxStra8Cre^* mice, Np expression was no longer detectable in spermatogonia and spermatocytes (Figure 6). In *Nptn^loxloxAmhCre^* mice, *Nptn* expression was ablated in Sertoli cells (Figure 6). In *Nptn^loxloxAmhCre+Stra8Cre^*, Np was completely absent from spermatocytes, spermatogonia, and Sertoli cells but still expressed by Leydig and myoid cells (Figure 6). The cell-type specific ablation of *Nptn* confirmed the neuroplastin expression revealed by our immunohistochemistry. *Nptn^loxloxStra8Cre^*, *Nptn^loxloxAmhCre^*, and *Nptn^loxloxAmhCre+Stra8Cre^* males sired offspring successfully, showing that Np expression by spermatocytes, spermatogonia, and Sertoli cells in the testis is not required for fertility or reproduction.

*Nptn^loxloxEmx1Cre^* males lacking Np expression, specifically in glutamatergic neurons [25], sired offspring similarly to wild-type males, excluding Np expression by glutamatergic neurons as essential for male fertility and mounting behavior. Similarly, male *Nptn^loxloxPrCreERT^* mice [6] that sired offspring before induction by tamoxifen injection continued to sire offspring for more than 2 months after inactivation of the *Nptn* gene, ruling out a role of neuronal Np expression for fertility and mounting behavior after completed development and maturation.

### 2.3. Hormonal Regulation in Nptn^−/−^ Male Mice

#### 2.3.1. Levels of Luteinizing Hormone (LH) and Follicle Stimulating Hormone (FSH) in *Nptn^−/−^* Male Mice

Elevated corticosterone levels observed in *Nptn^−/−^* mice [6] may inhibit the secretion of LH [26], which triggers the production of testosterone in the Leydig cells of the testis [27]. We determined LH and FSH levels in the blood but did not detect significant differences in the levels of LH (Figure 7A) and FSH (Figure 7B) in adult *Nptn^−/−^* male mice when compared to their wild-type *Nptn^+/+^* littermates.

#### 2.3.2. *Nptn^−/−^* Male Mice Do Not Attain Adult Levels of Testosterone

The intra-testicular levels of testosterone in adult *Nptn^−/−^* (*n* = 6) and littermate *Nptn^+/+^* (*n* = 6) male mice did not differ (Figure 7C), conforming to the observed normal development of testis and sperm in *Nptn^−/−^* males. Testosterone levels in the blood drastically increase at puberty and decline with age [28]. At 3 months of age, testosterone levels in *Nptn^+/+^* male mice were approximately 19-fold higher compared to juvenile 1-month-old *Nptn^+/+^* males (Figure 7D; *n* = 5; one-way ANOVA; F_(1,6)_ = 173.202; *p* < 0.0001). At the age of 11 months, *Nptn^+/+^* males displayed approximately three-fold lower levels of testosterone compared to the 3-month-old male mice (*n* = 3; one-way ANOVA; F_(1,4)_ = 30.797; *p* = 0.0052). In contrast, testosterone levels of *Nptn^−/−^* males examined at 1, 3, or 11 months of age were not significantly different from each other. When compared to their wild-type littermates, the levels of testosterone in the *Nptn^−/−^* male mice were found to be approximately 19-fold lower at 3 months (one-way ANOVA, F_(1,4)_ = 101.895, *p* = 0.0005) and approximately 6-fold lower at 11 months of age (one-way ANOVA, F_(1,4)_ = 7.936, *p* = 0.048).

#### 2.3.3. Reduced PMCA1 Expression in *Nptn^−/−^* Leydig Cells

Np is crucial for the expression of PMCAs [16,17,25], regulating Ca^2+^-efflux and potentially interfering with elevation of the intracellular Ca^2+^ level accompanied with LH-mediated stimulation of testosterone production by Leydig cells. Using immunohistochemistry, we observed a decrease in PMCA1 expression in the testis, particularly in the Leydig cells of *Nptn^−/−^* males compared to *Nptn^+/+^* littermates (Figure 7E). The decrease in PMCA1 expression in the testis could be confirmed by Western blot analysis (Figure 7F; testis *Nptn^+/+^* 3.013 ± 0.051, *Nptn^−/−^* 1.84 ± 0.314 relative units, 39% decrease, one-way ANOVA, F_(1,4)_ = 13.588, *p* = 0.0211; brain *Nptn^+/+^* 1.145 ± 0.11, *Nptn^−/−^* 0.617 ± 0.01 relative units, 46% decrease, one-way ANOVA, F_(1,4)_ = 22.716, *p* = 0.0089).

LH-receptor and PKA are critical components of LH signaling, and HSD3B1 (3ß-hydroxysteroid dehydrogenase/isomerase type 1) and CYP11A1 are the critical enzymes for testosterone synthesis. In adult testis homogenates, significant differences in expression of the regulatory subunit IIa of PKA (PKA[RIIa]), LH-receptor, HSD3B1, and CYP11A1 (Figure 8A) were not detected. However, the subcellular localization of PKA[RIIa] was altered from more membrane-associated in *Nptn^+/+^* to more cytoplasmic localization in *Nptn^−/−^* (Figure 8B) accompanied by a significant reduction in PKA[RIIa] staining intensity in testis from *Nptn^−/−^* compared to *Nptn^+/+^* male mice (Figure 8C; one-way ANOVA, *Nptn^−/−^ n* = 14 sections (132 tubules), *Nptn^+/+^ n* = 9 sections (100 tubules), F_(1,21)_ = 5.795, *p* = 0.0254). This finding may be particularly interesting because A-kinase anchoring protein (AKAP)-mediated multiprotein complex formation and the tethering of PKA holoenzyme via their regulatory subunit II dimers to defined cellular sites determines the efficiency and specificity of cAMP signaling (for review, see [29]).

## 3. Discussion

Genetic abnormalities are estimated to cause or contribute to about 15–30% of male infertility. However, the majority of genetic causes of male infertility are still unknown (for review, see [1,3]). The role of Np in reproduction was only recently suggested by the inability of Np-deficient male mice to sire offspring [6], which adds *Nptn* to the genes essential for fertility. Mice lacking the Np paralog basigin (EMMPRIN, CD147) display sterility of both sexes [21]. In female basigin-deficient mutants, the ovaries and genital tract are morphologically normal, but the uterus is probably incapable of implantation [21,30]. Basigin-deficient male mice are sterile due to the failure of spermatogenesis and show smaller testes compared to wild-type [21,22,31]. In this study, we addressed the novel role of Np for male fertility in detail. We confirmed that male *Nptn^−/−^* mice do not produce any offspring, whereas female *Nptn^−/−^* mice appear to be fully fertile. Np expression in the testis was confirmed by RT-PCR, Western blot, and immunohistochemistry, which revealed Np in the adult testis most prominently in Leydig, Sertoli, and myoid cells and in spermatogonia and spermatocytes. On mature isolated sperm, Np was detected on heads and tails. These data are in perfect agreement with the data from single-cell RNA-sequencing studies [24].

In *Nptn^−/−^* mice, the morphological development of genitalia and testes was found to be normal with respect to size, weight, and histology. The expression patterns of markers for Leydig cells (CPY11A), Sertoli cells (Sox9), spermatogonia, spermatocytes (Stra8), and myoid cells (alpha-smooth muscle actin) appeared normal in the adult *Nptn^−/−^* testis, supporting that a morphologically normal maturation of the testis proceeds also in the absence of Np. Furthermore, the conditional ablation of Np expression in spermatogonia, spermatocytes, and Sertoli cells using *Nptn^loxloxAmhCre+Stra8Cre^, Nptn^loxloxStra8Cre^*, and *Nptn^loxloxAmhCre^* mice did not interfere with reproduction, and males remained fully fertile and produced offspring. These results indicate that Np expression is not required for spermatogenesis and that Np expression in Sertoli cells, spermatogonia, and spermatocytes is not needed for reproduction. This is strikingly different from basigin expression, which is required on Sertoli cells for the maturation of spermatogonia to spermatocytes. Noteworthy is the pattern of Np staining on mature sperm, which was punctate, whereas basigin was more uniformly expressed along sperm heads and tails. For *Nptn^−/−^* male mice, the analysis of sperm development, maturation, number, and motility did not yield any indication of compromised spermatogenesis. Our observation of fully absent mounting behavior prompted us to consider a neurological base underlying infertility in *Nptn^−/−^* males. However, selective lack of Np from glutamatergic neurons, which express approximately 95% of brain Np, or inactivation of *Nptn* in the adult CNS did not interfere with reproduction of the males, ruling out a role of neuronal Np expression directly or indirectly via hormonal regulation for fertility and mounting behavior. Unfortunately, due to the lack of suitable Cre-deleter mouse strains, the conditional ablation of *Nptn* specifically only in Leydig or smooth muscle cells was not possible, leaving the possibility that Np expression by these cells is decisive.

Leydig cells present in the interstitial tissue of seminiferous tubules in the testes are the major site of testosterone production in mammals [32]. Testosterone synthesis in adult Leydig cells is triggered by LH, and the proliferation of Leydig cells depends on FSH acting on Sertoli cells; these two essential gonadotropins are secreted by the anterior pituitary gland [26,33]. Furthermore, FSH influences testis size and is required for normal Sertoli and germ cell numbers, and the lack of FSH reduces sperm motility [34,35]. FSH receptor knockout mice produce reduced testosterone levels, show reduced testis size, spermatogenesis, and sperm motility, and display aberrant sperm morphology and reduced fertility [36]. Complete lack of LH signaling as in LH-receptor knockout mice preventing postnatal, but not prenatal, Leydig cell maturation and androgen production leads to developmental and morphological deficits with resulting infertility that can, in part, be overcome by testosterone replacement [37]. Noteworthy, corticosterone at concentrations higher than physiological levels, as observed in *Nptn^−/−^* mice [6], can potentially inhibit the production of testosterone in Leydig cells [38]. Our analysis provided no hint for abnormal testis morphology or altered sperm capabilities of *Nptn^−/−^* males, arguing against LH- or FSH-deficits. Furthermore, we observed normal concentrations of LH and FSH in the blood of *Nptn^−/−^* males supporting a functional HPA axis. Interestingly, Oduwole et al. [39] described a patient with a partial selective defect of LH function caused by a homozygous LHß defect that partially affected LH signaling. The reduced intratesticular testosterone was sufficient to allow for normal spermatogenesis with complete germ cell maturation locally. The intratesticular testosterone concentration of this patient was locally sufficient for spermatogenesis but inadequate to maintain systemic T at levels required to induce virilization because of hormone dilution in blood and extracellular space. In *Nptn^−/−^* mice, the testosterone level in the testis itself was unaffected by the lack of Np; however, the expected increase in the blood level from puberty to adulthood was missing in *Nptn^−/−^* males. Unlike adult Leydig cells, fetal Leydig cells in the mouse do not require LH to stimulate androgen production and are capable of producing sufficient levels of androgens in the absence of LH stimulation to induce male fetal masculinization [26]. Furthermore, sexual differentiation occurs before maturation, which may explain why the intratesticular testosterone production in *Nptn^−/−^* suffices for testis development. Importantly, Leydig cell number and testis size develop normally in *Nptn^−/−^* whereas in mice lacking LH or the LH receptor, this is not the case [40,41,42], showing that the *Nptn^−/−^* phenotype is not identical to the complete lack of LH-signaling. In *Nptn^−/−^*, probably insufficient amounts of testosterone are produced and/or released, not fulfilling the large demand of constant testosterone in the bloodstream during adulthood. According to our measurements, the total amount of testosterone in the blood of male mice is higher than the total amount of testosterone in the testis itself. Furthermore, quick conversion of testosterone in the blood and liver leads to underestimation of testosterone secreted into the blood.

A potential explanation for the low testosterone level in the blood of *Nptn^−/−^* male mice could be a partially abnormal response to LH signal processing, which is associated with the elevation of intracellular [Ca^2+^] [43]. In Leydig cells, LH binding to the G-protein coupled LH receptor activates cAMP, producing adenylate cyclase and PKA and leading to intracellular [Ca^2+^] elevation [43]. Restoration of normal intracellular Ca^2+^ levels after signaling is achieved by PMCAs extruding Ca^2+^ to the extracellular space (for review, see [44]). Complex regulation of PMCAs via the cAMP/PKA/calmodulin signaling pathway has been reviewed [45], and PMCA1 is efficiently phosphorylated by PKA [46]. Np forms complexes with PMCA and is required for normal expression levels of PMCAs [6,11,16,17,25]. Therefore, we investigated PMCA levels in the testis and observed that PMCA1 levels in the testis, particularly in Leydig cells of *Nptn^−/−^* males, are lower compared to wild-type males. Furthermore, PKA[RIIa] localization was altered in Leydig cells. Similar to the altered Ca^2+^ extrusion kinetics that we observed for reduced PMCA in various cell types (e.g., neurons, T-cells [16,25]), reduction in PMCA1 levels in Leydig cells may lead to the alteration of intracellular Ca^2+^ dynamics and eventually insufficient response to LH signals. With reduced amounts of PMCA1 in the absence of Np, PKA-mediated activation of Ca^2+^ extrusion is compromised, potentially causing a block in further LH signaling. LH signaling, which increases intracellular Ca^2+^ to trigger testosterone production by Leydig cells, may be silenced by elevated intracellular Ca^2+^ levels and altered Ca^2+^ kinetics caused by reduced PMCA1 in the absence of Np. Therefore, we conclude that Np is required to obtain adult testosterone blood levels and to ensure successful mounting and male mating behavior. It will be interesting to investigate whether testosterone supplementation can rescue the mating behavior and fertility of *Nptn^−/−^* male mice.

## 4. Materials and Methods

### 4.1. Animals

Neuroplastin-deficient mice *Nptn^−/−^* (*Nptn^tm1.2Mtg^*) and floxed *Nptn^loxlox^* mice with neuron-specific inducible PrCreERT (*Nptn^loxloxPrCreERT^*) or with conditional lack of Np in Emx1-expressing cells (*Nptn^loxloxEmx1Cre^*) were described [6,25]. Mice lacking Np conditionally in Sertoli cells (*Nptn^loxloxAmhCre^*) or in spermatogonia (*Nptn^loxloxStra8Cre^*) were obtained by crossing floxed *Nptn^loxlox^* mice with 129S.FVB-Tg(Amh-cre)8815Reb/J (strain 007915, The Jackson Laboratory, Bar Harbor, ME, USA) or B6.FVB-Tg(Stra8-icre)1Reb/LguJ (strain 017490, The Jackson Laboratory), respectively, and further backcrossing to *Nptn^loxlox^* mice. Double Cre-mutants (*Nptn^loxloxAmhCre+Stra8Cre^*) lacking Np in Sertoli cells and in spermatogonia were obtained by intercrossing the single mutants.

Mice were kept with a 12 h light/dark cycle and food and water ad libitum. Animal husbandry and tissue collection for RNA, protein, and histochemical analysis were conducted in accordance with German (Tierschutzgesetz TierSchG) and European legislations (European Communities Council Directive (2010/63/EU) for the care of laboratory animals) and with the respective legal and ethical approval by the legal authorities (Landesverwaltungsamt Halle, Sachsen-Anhalt, Germany).

### 4.2. Breeding and Mating Analysis

To determine the fertility of males, 6 *Nptn^−/−^* males were tested. In the LIN breeding facility, each male 2–3-month-old mouse was paired with 2 female 8-week-old wild-type C57BL6/NCrl (Charles River Laboratories, Sulzfeld, Germany) mice in a standard cage (Greenline IVC, Tecniplast S.p.A., Buguggiate, Italy) for 2 months after which the females were replaced by 2 new 8-week-old females until retirement of the male at ≥1 year age. No offspring from any *Nptn^−/−^* male nor any pregnancy were obtained. As controls, *Nptn^+/−^* males were paired with *Nptn^+/−^* females or with C57BL/6NCrl females and yielded offspring in comparable numbers to wild-type *Nptn^+/+^* males. In each of the 6 breedings, 1 *Nptn^+/−^* male was crossed with 2 *Nptn^+/−^* females during 6 months, resulting in 45 litters with 272 offspring at weaning age (6 offspring/litter, 138 males, 134 females).

For mating behavior analysis, adult, socially naive male mice (6 *Nptn^−/−^* males and 5 *Nptn^+/+^* littermate males) were exposed to socially naive wild-type females for 30 min, and their behavior was recorded using an overhead camera and videotaped. The latency to and the number of mounts, mounts with thrusts, and the sniffing behavior (duration) were scored by inspection of the video records by an investigator unaware of the genotype.

### 4.3. Quantitative RT-PCR

After the dissection of adult male mice, total RNA was isolated from the fresh frozen tissues indicated in Figure 3, using the RNeasy total RNA prep according to the manufacturer’s instructions (Qiagen, Hilden, Germany) and eluted in a suitable volume of DEPC-treated water. The RNA concentration was determined by OD measurement using nanodrop. Only RNA samples with an A_260/280_ value greater than 2 were used for the subsequent real-time PCR experiments. Real-time PCR experiments were performed using Taqman gene expression assays. Every 1 μg total RNA was reverse-transcribed into cDNA using the high capacity RNA to cDNA Master mix (Applied Biosystems Thermo Fisher Scientific, Darmstadt, Germany). The PCR amplification was performed with the Applied Biosystems 7300 Real-Time PCR instrument with one-tenth of the first strand reaction using the Taqman gene expression Master mix. GAPDH was chosen as the endogenous control in all reactions. Relative quantification was performed using the 7300 version 1.4 software. Gene expression assays were obtained from Applied Biosystems Thermo Fisher Scientific. The Taqman gene expression assays were chosen to span exon borders (neuroplastin Mm00485993_m1 and Mm00485990_m1).

### 4.4. Western Blotting

The whole brain, testis, and sperm from male mice were homogenized in 50 mM Tris-HCl buffer (pH 8.1) with a protease inhibitor cocktail (Roche, Mannheim, Germany). The crude membranes were extracted by adding lysis buffer (50 mM Tris-HCl, 1% Triton X-100, and protease inhibitor cocktail, pH 8.1). Following incubation for 1 h on ice, the supernatant was collected by centrifugation at 12,000× *g* for 20 min. Samples were denatured with 2X SDS loading buffer for 5 min at 95 °C, separated on 10 or 15% SDS-PAGE-gels, and transferred by Western blot. Blots were incubated overnight with the first antibody. Primary antibodies used were sheep polyclonal anti-Np detecting Np65 and Np55 pan-Np55/65 and goat polyclonal isoform-specific anti-Np65, R&D systems, Wiesbaden, Germany; rabbit anti-PMCA1, rabbit anti-LH-receptor, and mouse monoclonal anti-ß-actin, Thermo Fisher Scientific, Dreieich, Germany; rabbit anti-alpha tubulin, Synaptic Systems, Goettingen, Germany; mouse monoclonal anti-PKARIIa, BD Transduction Laboratories, Heidelberg, Germany; rabbit anti-HSD3B1, Sigma, St. Louis, MO, USA; rabbit anti-CYP11A1, Abcam, Berlin, Germany; mouse monoclonal anti-GAPDH, Santa Cruz, Dallas, TX, USA. After incubation for 1 h with the secondary antibody (anti-sheep, -goat, -mouse, or -rabbit IgG antibody, Jackson ImmunoResearch Dianova, Hamburg, Germany) and washing with TBS containing 0.5% Tween 20, the membrane was developed with ECL solution (Intas Chemocan ECL Imaging, Göttingen, Germany). Staining for alpha-tubulin, ß-actin, or GAPDH served as loading control.

### 4.5. Enzyme-Linked Immunosorbent Assays

ELISAs were performed exactly according to the instructions provided by the manufacturer. Briefly, ELISA kits were purchased from TECAN IBL International (Hamburg, Germany) for testosterone (RE52151) and Follicle Stimulating Hormone (FSH, RE52121), and USCN life sciences (Aachen, Germany) for Luteinizing Hormone (LH) (E90441Mu). Testosterone, FSH, and LH were determined in the serum of blood collected by heart puncture. In addition, testosterone was analyzed in testis homogenates. In a well of an antibody-coated microtiter plate, 25 µL sample and 200 µL of enzyme conjugate were mixed and incubated for 60 min at room temperature on an orbital shaker. After discarding the incubation solution, the plate was washed 3 times with 300 µL of diluted wash buffer, and the excess buffer solution was removed. An amount of 100 µL of TMB substrate was placed into each well. After 15 min at room temperature, the subtract reaction was stopped by the addition of 100 µL TMB stop solution. After gentle shaking of the plate, the optical density was measured with a photometer at 450 nm.

### 4.6. Histological Examination of Testis Morphology

Testes were dissected from transcardially perfused males, fixed in Bouin’s fluid, and paraffin-embedded using a standard protocol. In short, they were subjected to immersion in increasing concentrations of ethanol with 70%, 80%, 90%, and 95%, and three changes of 100% for one hour each, subsequently immersed in four changes of xylene for two hours, and then, three changes of paraffin for three hours. Finally, they were embedded in paraffin and stored at 4 °C. An amount of 10 μm sections were obtained on gelatine-treated slides in a 50 °C water bath by a microtome. Standard Haematoxylin & Eosin staining was performed, and sections were viewed using an Axioplan2 (Zeiss, Jena, Germany) microscope. Brightfield images were obtained using 2.5× magnification, and the number and perimeter of seminiferous tubules per transverse section were registered using ImageJ (Version 2.9.0/1.5.3t).

### 4.7. Immunofluorescence Staining after Paraffin Embedding

Paraffin-embedded testis sections were deparaffinized and blocked with BSA (5% in phosphate-buffered saline) for 1 h and then incubated with primary antibodies (sheep polyclonal Np detecting Np65 and Np55 (pan-Np55/65; 1:500, R&D systems) overnight at 4 °C in a humidified chamber. Post incubation, the sections were washed with PBS and probed with the secondary antibody (Cy3-conjugated anti-sheep 1:1000, Jackson Immunoresearch) in a sequential manner. Sections were examined by an Axioplan2 (Zeiss), and images were captured by Spot RT-KE camera (Diagnostic Instruments, Marburg, Germany).

### 4.8. Immunofluorescence Staining after Paraformaldehyde Fixation

Adult mice were anesthetized with isoflurane and transcardially perfused with PBS, followed by 4% PFA. Testes were dissected and post-fixed in 4% PFA overnight. Testis sections were blocked (20% horse serum in PBS, one hour, room temperature) and incubated with primary antibodies in PBS containing 0.3% Triton X-100 and 10% horse serum (overnight, 4 °C). Primary antibodies used were sheep polyclonal anti-Np detecting Np65 and Np55 (pan-Np55/65; 1:300, R&D systems) and rabbit polyclonal anti-Stra8 (Stimulated By Retinoic Acid 8, used as a marker for spermatogonia and spermatocytes, 1:500, Abcam), anti-CPY11A1 (Cytochrome P450 Family 11 Subfamily A Member 1, marker for Leydig cells, 1:500, Abcam), anti-alpha-smooth muscle actin (marker for myoid cells, 1:500, Abcam), and anti-Sox9 (SRY-Box Transcription Factor 9, marker for Sertoli cells, 1:500, EMD Millipore, Darmstadt, Germany) antibodies. The specificity of antibodies was assessed with negative controls using pre-immune sera.

Secondary antibodies were Cy3-conjugated anti-sheep and Cy5-conjugated anti-rabbit (1:1000, Jackson Immunoresearch). After washing with PBS and briefly with water, the sections were mounted on glass slides with fluoromount g DAPI (Southern Biotech, Birmingham, AL, USA) and visualized using a Leica SP5 confocal microscope (Leica, Wetzlar, Germany).

### 4.9. Epididymal Sperm Motility and Ploidy Stages of Testis Cells

Epididymal sperm was prepared as previously described [47]. Excised and cleaned cauda epididymidis and vas deferens were washed, transferred to 1 mL buffer HS (in mM: 135 NaCl, 5 KCl, 2 CaCl_2_, 1 MgCl_2_, 20 HEPES, 5 glucose, 10 DL-lactic acid, and 10 pyruvic acid pH 7.4) and incised 3–5 times to allow the sperm to swim out into the medium over a period of 20 min at 37 °C and under 5% CO_2_. After washing three times in buffer HS, sperm was resuspended in 0.5 mL buffer HS at a final concentration of 1–2 × 10^7^ cells/mL and used for determination of motility. The beat frequency of sperm was measured as described in [47] before and after local perfusion of the chamber with bicarbonate (15 mM; HSB), which activates a soluble adenylate cyclase, increasing the beat frequency.

For the analysis of spermatogenetic activity according to [48], frozen testis parenchyma was thawed, dissociated, and testicular cells were applied to flow cytometric DNA analysis: about 50 mg parenchyma was finely minced in 0.8 mL 100 mM citric acid containing 0.5% (*v*/*v*) Tween 20 and agitated for 20 min at room temperature. The DNA of lysed cells was stained by adding 4 mL of a 400 mM Na_2_HPO_4_ solution containing 5 µM 4′,6-diamidino-2-phenylindol (DAPI) for 10 min in the dark. Measurements were performed on a PAS III flow cytometer (Sysmex Deutschland GmbH, Norderstedt, Germany) equipped with a UV LED and an appropriate filter set (excitation: 365 nm, emission: 420 nm). Cells were recorded for their fluorescence intensity corresponding to the DNA content of their nuclei, and the histograms were analyzed for the proportions of cells in each peak by the FlowMax version 2.3 software (Sysmex). Haploid signals (1c) come from postmeiotic germ cell stages like spermatids and sperm cells, diploid signals (2c) from spermatogonia, secondary spermatocytes, and somatic testicular cells, tetraploid signals (4c) mainly derive from meiotic primary spermatocytes but also in mitotic spermatogonia. Between 2c and 4c, cells in the S-phase (S) were detected.

### 4.10. Immunofluorescence Staining of Sperm

Sperm was stained using sheep polyclonal antibodies against Np detecting Np65 and Np55 (pan-Np55/65 1:500, R&D systems). Secondary antibodies were Cy3-conjugated anti-sheep (1:1000, Jackson Immunoresearch). Counterstaining used phalloidin-iFluor 488 green (1:1000, Abcam, Berlin, Germany) and fluoromount g DAPI (Southern Biotech). Immunofluorescence was visualized using a Leica SP5 confocal microscope.

### 4.11. Statistical Analysis

Statview 5.0.1 (SAS Institute, Inc., Cary, NC, USA) was used for the analysis of variance (ANOVA), post hoc analysis (Scheffé or Fisher’s protected least significant difference), repeated-measures analysis of variance, and *t*-tests. GraphPad Prism 9.4.1 (San Diego, CA, USA) was used for *t*-tests and graph design. *p* < 0.05 was considered significant. For all analyses in this manuscript, all data points were included, and outliers were not removed.

## Figures and Tables

**Figure 1 ijms-25-00177-f001:**
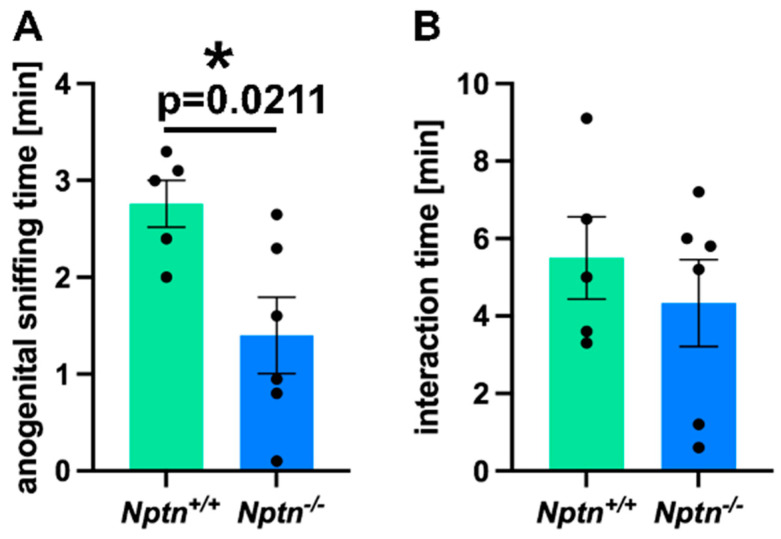
Mating behavior of *Nptn^−/−^* male mice. (**A**) The time spent with anogenital sniffing at socially naive wild-type females was significantly shorter (*p* = 0.0211) for *Nptn*^−/−^ (*n* = 6) compared to *Nptn^+/+^* (*n* = 5) adult, socially naive male mice during 30 min exposure (one-way ANOVA, * *p* < 0.05). (**B**) The interaction time with socially naive wild-type females was similar for *Nptn^+/+^* (*n* = 5) and *Nptn^−/−^* (*n* = 6) adult, socially naive male mice during 30 min exposure (one-way ANOVA).

**Figure 2 ijms-25-00177-f002:**
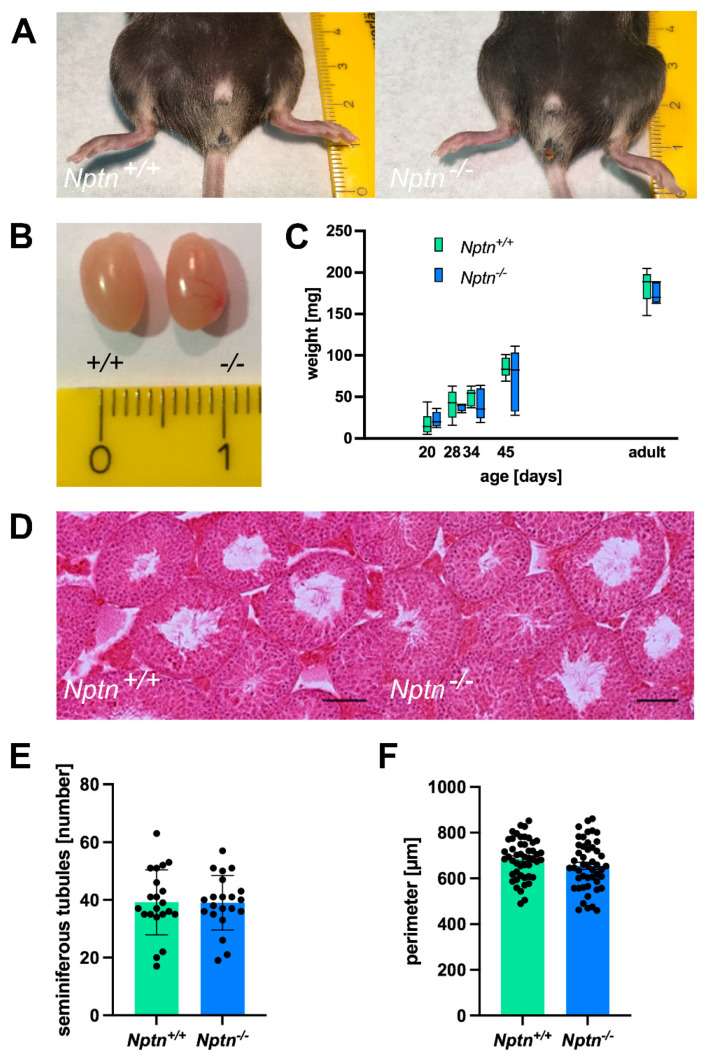
Normal appearance of genitalia and testes of *Nptn^−/−^* male mice. (**A**) Outer genitalia of *Nptn^−/−^* male mice are normally developed (ruler in A and B: cm). (**B**) The testes of *Nptn^−/−^* male mice have normal morphology and size. (**C**) The average weights of testes from *Nptn^−/−^* and *Nptn^+/+^* male littermates are similar (one-way ANOVA, *n* = 5, F (1,8) = 0.726, *p* = 0.419). Boxes indicate minimum-to-maximum range of data; for individual data points, see Appendix A. (**D**) The gross morphology of *Nptn^+/+^* and *Nptn^−/−^* testes was found to be comparable, as revealed by standard Haematoxylin & Eosin staining (scale bar = 500 μm). (**E**) The average number of fully formed and identifiable seminiferous tubules counted per testis transverse section was not significantly different between the genotypes (7 random sections from each animal; *n* = 3 mice for each genotype were selected for counting). (**F**) The average perimeter of seminiferous tubules was similar in *Nptn*^+/+^ and *Nptn*^−/−^ male mice.

**Figure 3 ijms-25-00177-f003:**
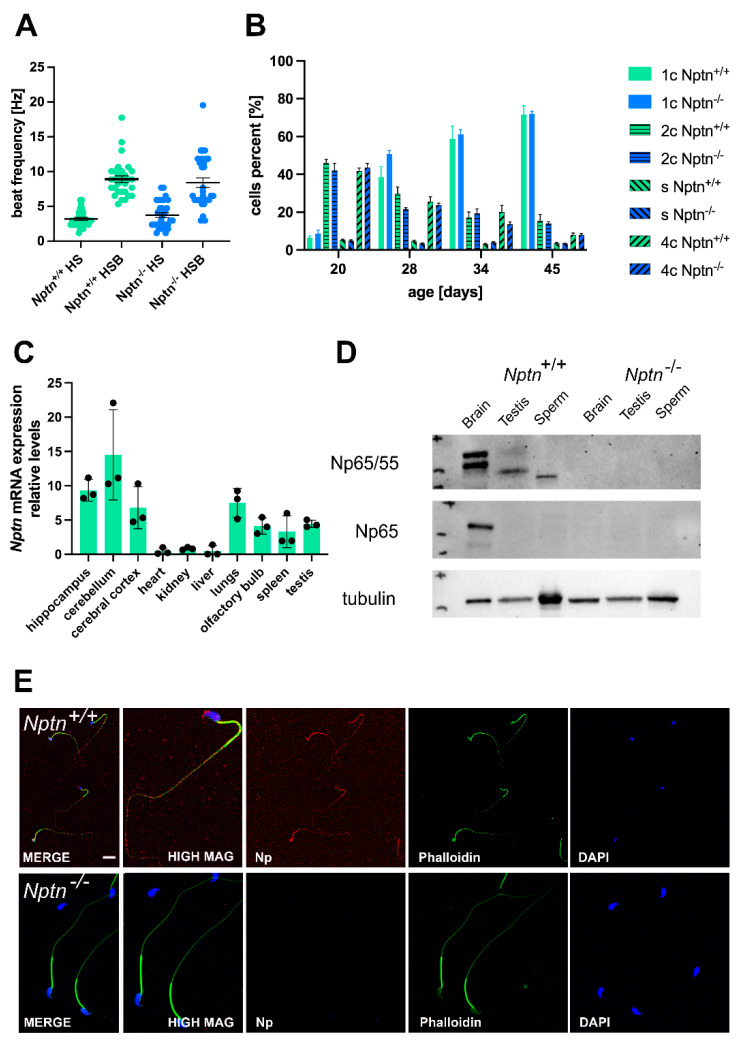
Neuroplastin expression in testis. (**A**) The beat frequency of sperm isolated from adult *Nptn^+/+^* and *Nptn*^−/−^ male mice was similar and could be increased by bicarbonate activation. (**B**) The ploidy status of testicular cells from male mice at different ages during maturation was examined. At 20, 28, 34, and 45 days of age, the proportion of haploid (1c), diploid (2c), and tetraploid (4c) cells from *Nptn*^+/+^ (*n* = 6 for each age) and *Nptn*^−/−^ (*n* = 4 for 20 and 28 days each; *n* = 6 for 34 and 45 days each) male mice did not differ. For individual data points, see Appendix A. (**C**) RT-PCR revealed that *Nptn* mRNA is expressed predominantly in the central nervous system but also in other organs, including the testis of *Nptn*^+/+^ male mice. (**D**) By Western blot analysis, Np expression was detected in testis and sperm of *Nptn^+/+^* but not in *Nptn^−/−^* male mice. The lower molecular weight reflects different glycosylation of Np in testis (50 kDa) and sperm (45 kDa) compared to brain (55 and 65 kDa). The neuron-specific isoform Np65 could not be detected in testis or sperm of *Nptn^+/+^* male mice (for original blots, see Appendix A). (**E**) Np (red) is expressed on isolated mature sperm heads and tails of *Nptn*^+/+^ male mice (upper panel), as detected by immunofluorescence (scale bar = 20 μm). The lower panel confirms lack of Np on sperm from *Nptn*^−/−^ male mice. Counterstaining was performed by phalloidin (green) and DAPI (blue).

**Figure 4 ijms-25-00177-f004:**
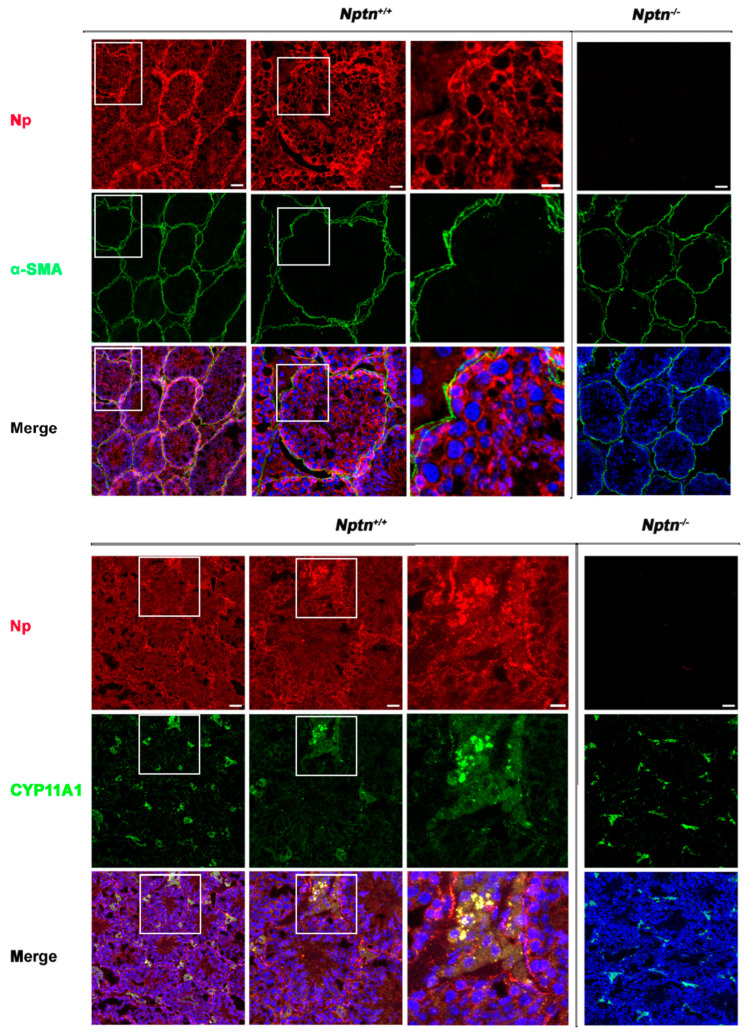
Neuroplastin expression by Leydig and peritubular myoid cells. Neuroplastin expression in the testis is clearly revealed by antibodies recognizing all neuroplastin isoforms (Np, red). In *Nptn^−/−^* testes, Np could not be detected. Antibodies against alpha-smooth muscle actin (⍺-SMA, green upper panel) identified peritubular myoid cells and demonstrated that Np was expressed by myoid cells in *Nptn^+/+^* (see also Figure 6 Amh+Stra8Cre) but not in *Nptn^−/−^*. Antibodies against CPY11A (green lower panel) identified Leydig cells and demonstrated that Np was expressed by Leydig cells in *Nptn^+/+^* but not in *Nptn^−^*^/−^. Note that expression of alpha-smooth muscle actin and of CPY11A are not altered in *Nptn^−/−^* (for *Nptn^+/+^* from left to right increasing magnification of boxed area; scale bar left = 50 µm; middle = 20 µm; right = 10 µm; for *Nptn^−^*^/−^, scale bar = 50 µm). In Merge, nuclei are labeled by DAPI (blue).

**Figure 5 ijms-25-00177-f005:**
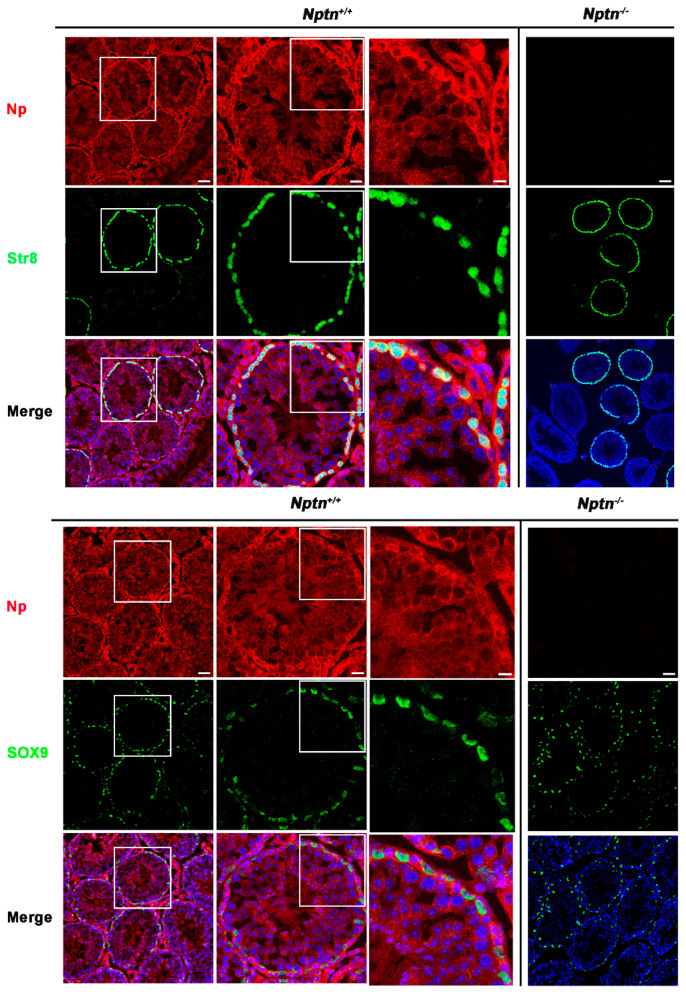
Neuroplastin expression by spermatogonia, spermatocytes, and Sertoli cells. Antibodies against Stra8 (green upper panel) identified spermatogonia and spermatocytes and demonstrated that Np (red) is expressed by spermatogonia and spermatocytes in *Nptn^+/+^* but not Nptn^−/−^. Antibodies against Sox9 (green lower panel) identified Sertoli cells and demonstrated that Np was expressed by Sertoli cells in *Nptn^+/+^* but not in *Nptn^−/−^*. Note that expressions of Stra8 and of Sox9 are not altered in *Nptn^−/−^* (for *Nptn^+/+^* from left to right increasing magnification of boxed area; scale bar left = 50 µm; middle = 20 µm; right = 10 µm; for *Nptn^−^*^/−^, scale bar = 50 µm). In Merge, nuclei are labeled by DAPI (blue).

**Figure 6 ijms-25-00177-f006:**
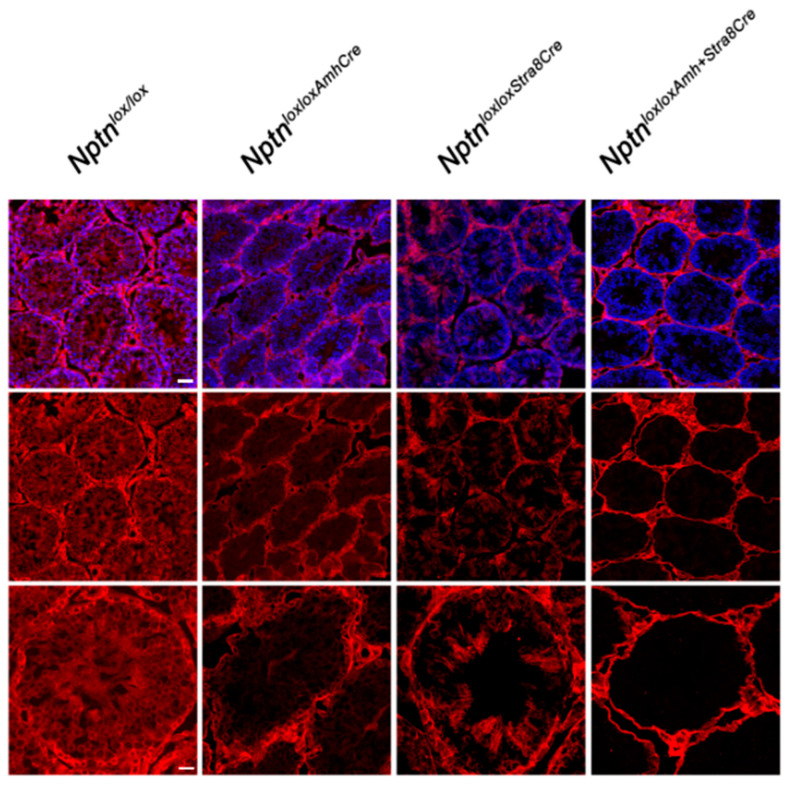
Conditional inactivation of *Nptn*. Cell type-restricted inactivation of the floxed *Nptn* gene (*Nptn^loxlox^*) was achieved using Amh-promoter (Anti-Mullerian Hormone, Sertoli cells) (*Nptn^loxloxAmhCre^*) or Stra8-promoter (*Nptn^loxloxStra8Cre^*) driven Cre, and double Cre (*Nptn^loxloxAmhCre+Stra8Cre^*) mice. Np was detected by antibodies recognizing all isoforms (red), and sections were counterstained with DAPI (blue; top panel; scale bar = 50 µm). Bottom panel shows higher magnification (scale bar = 20 µm).

**Figure 7 ijms-25-00177-f007:**
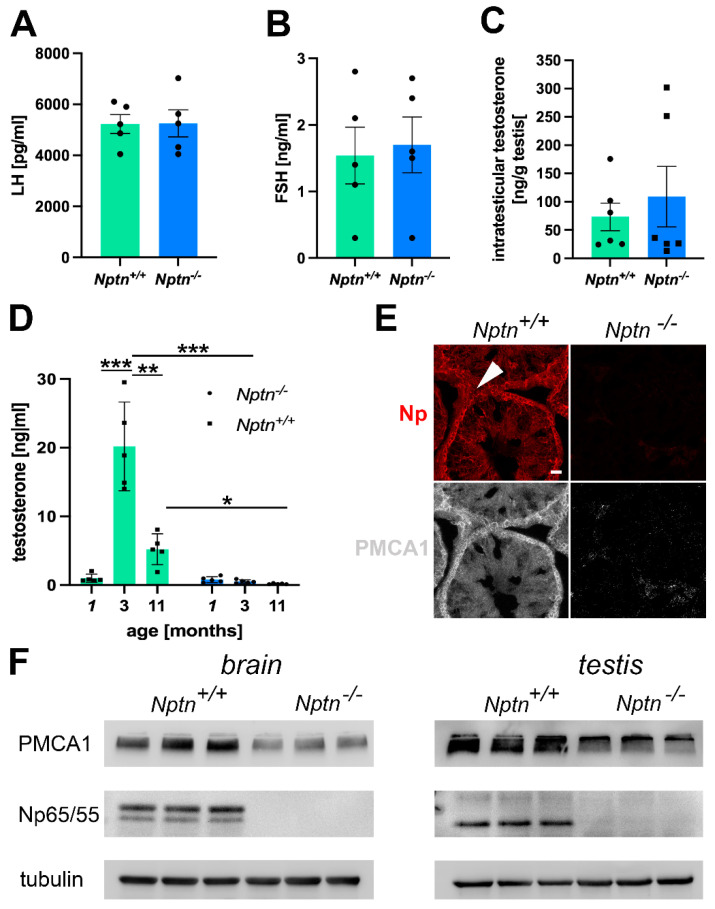
Hormonal status in *Nptn^−/−^* male mice. (**A**) LH levels, as determined by ELISA in the blood of adult *Nptn^+/+^* (*n* = 5) and *Nptn*^−/−^ (*n* = 5) male mice, were similar (one-way ANOVA). (**B**) FSH levels, as determined by ELISA in the blood of adult *Nptn^+/+^* (*n* = 5) and *Nptn^−/^*^−^ (*n* = 5) male mice, were similar (one-way ANOVA). (**C**) Intratesticular testosterone levels determined by ELISA in adult *Nptn^+/+^* (*n* = 6) and *Nptn^−/−^* (*n* = 6) male mice were similar (one-way ANOVA). (**D**) Testosterone levels, as determined by ELISA in the blood of juvenile (1-month-old; *n* = 5), adult (3-month-old; *n* = 5), and aged (11-month-old; *n* = 5) *Nptn^+/+^* male mice differed significantly, showing the strong increase after puberty and decline with age. In *Nptn^−/−^* male mice, low juvenile (*n* = 5) testosterone levels did not increase with age and were significantly lower in adult (*n* = 5) and aged (*n* = 5) males compared to *Nptn^+/+^* (* *p* < 0.05, ** *p* < 0.01, *** *p* < 0.001; one-way ANOVA). (**E**) Reduced expression of PMCA1 in testes of *Nptn^−/−^* compared to *Nptn^+/+^* male mice was revealed by immunohistochemistry (arrowhead points to a Leydig cell; scale bar = 20 µm). (**F**) Reduced expression of PMCA1 in brain (46% decrease) and testis (39% decrease) of adult *Nptn^−/−^* (*n* = 3) compared to *Nptn^+/+^* (*n* = 3) male mice was revealed by Western blot analysis (one-way ANOVA) (for original blots, see Appendix A).

**Figure 8 ijms-25-00177-f008:**
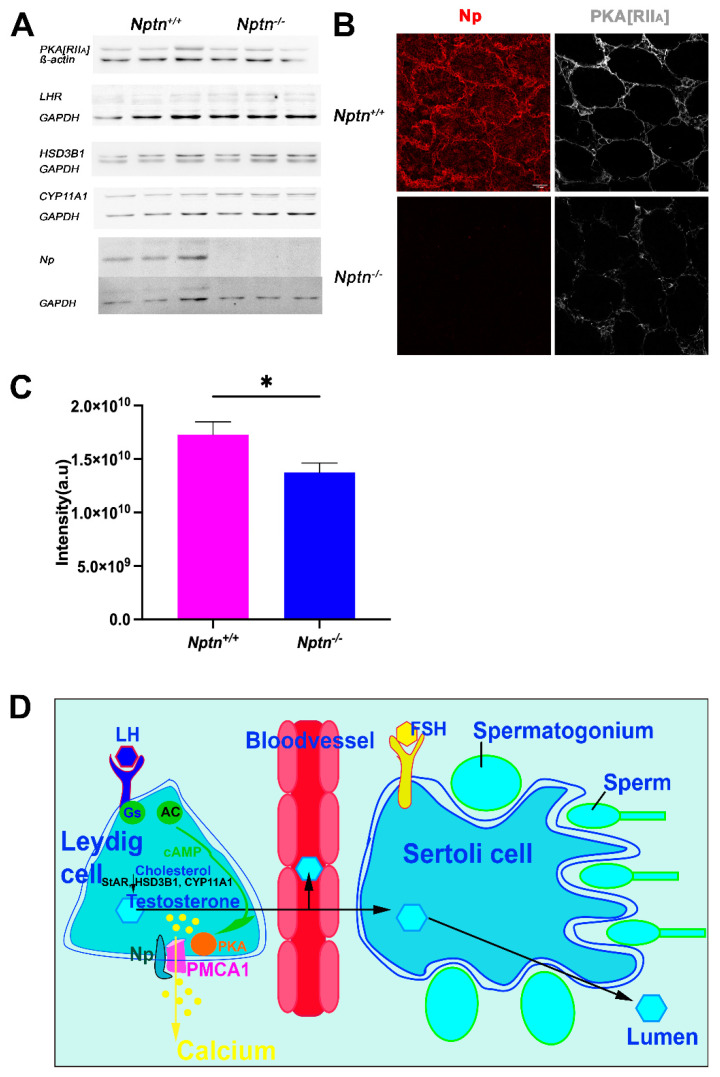
Expression of LH-Receptor, PKA[RIIa], and critical enzymes for testosterone synthesis in *Nptn^−/−^* male mice. (**A**) Expressions of PKA[RIIa], LH-receptor (LHR), HSD3B1, and CYP11A1 in testis of adult *Nptn^+/+^* (*n* = 3) compared to *Nptn^−/−^* (*n* = 3) male mice was analyzed by Western blot and did not reveal significant quantitative differences with respect to genotype (one-way ANOVA) (for original blots, see Appendix A). Np and GAPDH served as controls. (**B**) Altered expression of PKA[RIIa] in testes of *Nptn^−/−^* compared to *Nptn^+/+^* male mice was revealed by immunohistochemistry (scale bar = 50 µm). Labeling for PKA[RIIa] in *Nptn^+/+^* Leydig cells was more associated with the cell membrane, whereas in *Nptn^−/−^*, it was more diffuse and cytoplasmic. (**C**) The fluorescence intensity of the immunohistochemical staining for PKA[RIIa] was measured and revealed a significant difference (one-way ANOVA, *Nptn^−/−^ n* = 14 sections (132 tubules), *Nptn^+/+^ n* = 9 sections (100 tubules), F_(1,21)_ = 5.795, *p* = 0.0254, * *p* < 0.05) between testis from *Nptn^−/−^* compared to *Nptn^+/+^* male mice. (**D**) Schematic illustration of testosterone release by Leydig cells in the testis. Whereas fetal Leydig cells do not require LH for testosterone production, LH binding to the G-protein (Gs) coupled LH receptor triggers cAMP production by Adenylate cyclase (AC), and an increase in intracellular calcium levels leads to testosterone production in adult Leydig cells. PMCA1 is a target of PKA, which is activated by increased cAMP levels. Calcium is extruded by PMCA1, which is stabilized by Np. Lack of Np might interfere with LH signaling, resulting in insufficient trigger of testosterone synthesis to obtain high levels in blood. Expression of StAR (Steroidogenic acute regulatory protein), HSD3B1, and CYP11A mediating conversion of cholesterol to testosterone are influenced by cAMP/PKA/CREB regulation.

## Data Availability

All data generated or analyzed during this study are included in this published article (and its Appendix A). Further data may be obtained from the corresponding author upon reasonable request.

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
