# Peer review of "Neuroplastin Expression in Male Mice Is Essential for Fertility, Mating, and Adult Testosterone Levels"

_ijms, 2023, doi:10.3390/ijms25010177_

Round 1
Reviewer 1 Report
Comments and Suggestions for Authors
The story presented by MS is very interesting. I have only three concerns regarding current edition.
1. How to prove Np could form the complexes with PMCA. The COIP experiment might provide the evidence on the relationship of PMCA with Np in the testes.
2. Generally, free testosterone rather than total testosterone exerts the regulatory roles of behaviors. So, the authors should detect the free testosterone levels to show solid evidence.
3. If this phenotype comes from the dysfunction of Leydig cells, is there significant difference in the cell number and morphology of Leydig cells?
Comments on the Quality of English Language
No comment to editor.
Author Response
Reviewer1
The story presented by MS is very interesting. I have only three concerns regarding current edition.
- How to prove Np could form the complexes with PMCA. The COIP experiment might provide the evidence on the relationship of PMCA with Np in the testes.
Np has been identified as auxilliary obligatory subunit of native PMCAs (Schmidt et al.) and the structure of Np-PMCA1 complexes has been determined (Gong et al.). The work of Gong et al., Schmidt et al., and us (citations below) revealed thatthe protein levels of all four PMCA paralogs (PMCA1-4) strongly depend on the binding to and the formation of stable protein complexes with Np.
Deshun Gong, Ximin Chi, Kang Ren, Gaoxingyu Huang, Gewei Zhou, Nieng Yan, Jianlin Lei, Qiang Zhou Structure of the human plasma membrane Ca2+-ATPase 1 in complex with its obligatory subunit neuroplastin Nat Commun. 2018 Sep 6;9(1):3623. doi: 10.1038/s41467-018-06075-7. PMID: 30190470 PMCID: PMC6127144 DOI: 10.1038/s41467-018-06075-7
Plasma membrane Ca2+-ATPases (PMCAs) are key regulators of global Ca2+ homeostasis and local intracellular Ca2+ dynamics. Recently, Neuroplastin (NPTN) and basigin were identified as previously unrecognized obligatory subunits of PMCAs that dramatically increase the efficiency of PMCA-mediated Ca2+ clearance. Here, we report the cryo-EM structure of human PMCA1 (hPMCA1) in complex with NPTN at a resolution of 4.1 Å for the overall structure and 3.9 Å for the transmembrane domain. The single transmembrane helix of NPTN interacts with the TM8-9-linker and TM10 of hPMCA1. The subunits are required for the hPMCA1 functional activity. The NPTN-bound hPMCA1 closely resembles the E1-Mg2+ structure of endo(sarco)plasmic reticulum Ca2+ ATPase and the Ca2+ site is exposed through a large open cytoplasmic pathway. This structure provides insight into how the subunits bind to the PMCAs and serves as an important basis for understanding the functional mechanisms of this essential calcium pump family.
Nadine Schmidt, Astrid Kollewe, Cristina E Constantin, Sebastian Henrich, Andreas Ritzau-Jost, Wolfgang Bildl, Anja Saalbach, Stefan Hallermann, Akos Kulik, Bernd Fakler, Uwe Schulte Neuroplastin and Basigin Are Essential Auxiliary Subunits of Plasma Membrane Ca2+-ATPases and Key Regulators of Ca2+ Clearance. Neuron. 2017 Nov 15;96(4):827-838.e9. doi: 10.1016/j.neuron.2017.09.038. Epub 2017 Oct 19. PMID: 29056295 DOI: 10.1016/j.neuron.2017.09.038
Plasma membrane Ca2+-ATPases (PMCAs), a family of P-type ATPases, extrude Ca2+ ions from the cytosol to the extracellular space and are considered to be key regulators of Ca2+ signaling. Here we show by functional proteomics that native PMCAs are heteromeric complexes that are assembled from two pore-forming PMCA1-4 subunits and two of the single-span membrane proteins, either neuroplastin or basigin. Contribution of the two Ig domain-containing proteins varies among different types of cells and along postnatal development. Complex formation of neuroplastin or basigin with PMCAs1-4 occurs in the endoplasmic reticulum and is obligatory for stability of the PMCA proteins and for delivery of PMCA complexes to the surface membrane. Knockout and (over)-expression of both neuroplastin and basigin profoundly affect the time course of PMCA-mediated Ca2+ transport, as well as submembraneous Ca2+ concentrations under steady-state conditions. Together, these results establish neuroplastin and basigin as obligatory auxiliary subunits of native PMCAs and key regulators of intracellular Ca2+ concentration.
Lin, X., Brunk, M. G. K., Yuanxiang, P., Curran, A. W., Zhang, E., Stober, F., Goldschmidt, J., Gundelfinger, E. D., Vollmer, M., Happel, M. F. K., Herrera-Molina, R., and Montag, D. (2021) Neuroplastin expression is essential for hearing and hair cell PMCA expression. Brain Struct Funct 226, 1533-1551
Korthals, M., Langnaese, K., Smalla, K. H., Kahne, T., Herrera-Molina, R., Handschuh, J., Lehmann, A. C., Mamula, D., Naumann, M., Seidenbecher, C., Zuschratter, W., Tedford, K., Gundelfinger, E. D., Montag, D., Fischer, K. D., and Thomas, U. (2017) A complex of Neuroplastin and Plasma Membrane Ca(2+) ATPase controls T cell activation. Sci Rep 7, 8358
Herrera-Molina, R., Mlinac-Jerkovic, K., Ilic, K., Stober, F., Vemula, S. K., Sandoval, M., Milosevic, N. J., Simic, G., Smalla, K. H., Goldschmidt, J., Bognar, S. K., and Montag, D. (2017) Neuroplastin deletion in glutamatergic neurons impairs selective brain functions and calcium regulation: implication for cognitive deterioration. Sci Rep 7, 7273
Bhattacharya, S., Herrera-Molina, R., Sabanov, V., Ahmed, T., Iscru, E., Stober, F., Richter, K., Fischer, K. D., Angenstein, F., Goldschmidt, J., Beesley, P. W., Balschun, D., Smalla, K. H., Gundelfinger, E. D., and Montag, D. (2017) Genetically Induced Retrograde Amnesia of Associative Memories After Neuroplastin Ablation. Biol Psychiatry 81, 124-135
Malci, A., Lin, X., Sandoval, R., Gundelfinger, E. D., Naumann, M., Seidenbecher, C. I., and Herrera-Molina, R. (2022) Ca(2+) signaling in postsynaptic neurons: Neuroplastin-65 regulates the interplay between plasma membrane Ca(2+) ATPases and ionotropic glutamate receptors. Cell Calcium 106, 102623
- Generally, free testosterone rather than total testosterone exerts the regulatory roles of behaviors. So, the authors should detect the free testosterone levels to show solid evidence.
1-2% of total testosterone in the blood are not bound to SHBG or albumin and considered to be the biologically active fraction. The exact direct measurement of free tesosterone requires sophisticated methods as equilibrium dialysis, centrifugal ultrafiltration, steady-state gel filtration, flow dialysis, and direct tracer analog immunoassays. Because of the strict correlation of free testosterone with total testosterone, the amount of free testosterone can and is usually calculated from total testosterone, SHBG (40 nmol/l), and albumin (4.3 g/dL) concentrations as detailed in J Clin Endocrinol Metab 84:3666-3672, 1999. Unless there is an alteration in the level of SHBG or albumin, the reduced total testosterone level observed in adult 3 months-old Nptn-/- males (10% of WT) corresponds to a reduced free testosterone level of 0.03376ng/ml = 1.69% compared to 0.53381 ng/ml = 2.67% in Nptn+/+ controls.
- If this phenotype comes from the dysfunction of Leydig cells, is there significant difference in the cell number and morphology of Leydig cells?
The number and morphology of Leydig cells is not altered in Nptn-/-. We did not observe any differences in Leydig cell number during quantification of the seminiferous tubules nor after staining with the Leydig cell marker CYP11A. Furthermore, we did not detect alterations during the ploidy count which could have indicated differences in Leydig cell number as described by Michaelis et al..
Marten Michaelis, Alexander Sobczak, Carolin Ludwig, Hana Marvanová, Martina Langhammer, Jennifer Schön, Joachim M Weitzel. Altered testicular cell type composition in males of two outbred mouse lines selected for high fertility. Andrology. 2020 Sep;8(5):1419-1427. doi: 10.1111/andr.12802. Epub 2020 May 16.
Reviewer 2 Report
Comments and Suggestions for Authors
In the manuscript ijms-2725420 is studied the reproductive performance of Nptn−/− male mice by means of histological, hormonal, and behavioral analyses. The study is well-designed and comprehensible. However, certain shortcomings have been found, especially in the Materials and Methods, and some aspects of the manuscript can be improved after revisions, as outlined below.
Abstract: The abstract should be revised according to the instructions for authors https://www.mdpi.com/journal/ijms/instructions
Introduction: In the first paragraph, regarding male infertility do you refer to men?
Line 55: please explain the abbreviation Np.
In the last paragraph in the Introduction section, you should mention the objectives and the hypothesis of the study and highlight the main conclusions.
Results: please move the lines 89-90 to the Materials and Methods, as well as anything related to Materials and Methods (see below). Additionally, remove the rationale for each analysis from the Results and move it to the Materials and Methods or Introduction i.e., 114-117, 166, 211-215, 229-236, 244-248, 267-274, 286-290, 297-299. Only, keep a brief interpretation of the results and the experimental conclusions in the Results section.
Discussion: the Discussion is well written and comprehensive with a thorough interpretation of the study's findings.
Materials and Methods: According to the Instructions for authors, Materials and Methods should be described with sufficient detail to allow others to replicate and build on published results. What tissues did you collected and how (line 446)? Please write the procedure (i.e., lines 524-525) or add a reference for this. Also, add the type of tissues collected (i.e., lines 480, 500) and for what assay. In the sub-sections of RT-PCR and ELISA you did not write the tissues/samples assayed.
Overall, the Materials and Methods lack details about the experimental design, which are presented in Results section. Please move anything related to experimental design from each Results sub-section (e.g., lines 88-95, 115-117, 120-121, 130-134, etc) into the Materials and Methods section e.g., number of participated mice, duration of experiment.
Statistical analysis: Please write in which cases the analysis of variance (ANOVA), post hoc analysis (Scheffé or Fisher’s protected least significant difference), repeated-measures analysis of variance, and t-tests were performed, for each case/assay separately.
Author Response
Reviewer 2
In the manuscript ijms-2725420 is studied the reproductive performance of Nptn−/− male mice by means of histological, hormonal, and behavioral analyses. The study is well-designed and comprehensible. However, certain shortcomings have been found, especially in the Materials and Methods, and some aspects of the manuscript can be improved after revisions, as outlined below.
Abstract: The abstract should be revised according to the instructions for authors https://www.mdpi.com/journal/ijms/instructions
We revised the abstract according to the instructions
Introduction: In the first paragraph, regarding male infertility do you refer to men?
Yes, to clearly indicate this "in human" was added in the 2nd sentence.
Line 55: please explain the abbreviation Np.
full name neuroplastin was added
In the last paragraph in the Introduction section, you should mention the objectives and the hypothesis of the study and highlight the main conclusions.
This paragraph was revised accordingly.
Results: please move the lines 89-90 to the Materials and Methods, as well as anything related to Materials and Methods (see below). Additionally, remove the rationale for each analysis from the Results and move it to the Materials and Methods or Introduction i.e., 114-117, 166, 211-215, 229-236, 244-248, 267-274, 286-290, 297-299. Only, keep a brief interpretation of the results and the experimental conclusions in the Results section.
The respective sentences were revised accordingly.
Discussion: the Discussion is well written and comprehensive with a thorough interpretation of the study's findings.
Materials and Methods: According to the Instructions for authors, Materials and Methods should be described with sufficient detail to allow others to replicate and build on published results. What tissues did you collected and how (line 446)? Please write the procedure (i.e., lines 524-525) or add a reference for this. Also, add the type of tissues collected (i.e., lines 480, 500) and for what assay. In the sub-sections of RT-PCR and ELISA you did not write the tissues/samples assayed.
We revised RT-PCR in MM
"After dissection of adult male mice, total RNA was isolated from the fresh frozen tissues indicated in Figure 3, ..."
The tissues analyzed by RT-PCR are indicated in Fig 3C. In Figure 3 the label C was added.
The ELISAs are now described in more detail in M&M:
Testosterone, FSH, and LH were determined in the serum of blood collected by heart puncture. In addition, testosterone was analyzed in testis homogenates. In a well of an antibody coated microtiter plate, 25µl sample and 200µl of enzyme conjugate were mixed and incubated for 60 min. at room temperature on an orbital shaker. After discarding the incubation solution, the plate was washed 3 times with 300µl of diluted wash buffer and excess buffer solution was removed. 100µl of TMB substrate was placed into each well. After 15 minutes at room temperature, the subtract reaction was stopped by addition of 100µl TMB stop solution. After gentle shaking of the plate, the optical density was measured with a photometer at 450nm.
Overall, the Materials and Methods lack details about the experimental design, which are presented in Results section. Please move anything related to experimental design from each Results sub-section (e.g., lines 88-95, 115-117, 120-121, 130-134, etc) into the Materials and Methods section e.g., number of participated mice, duration of experiment.
The requested changes have been done.
Statistical analysis: Please write in which cases the analysis of variance (ANOVA), post hoc analysis (Scheffé or Fisher’s protected least significant difference), repeated-measures analysis of variance, and t-tests were performed, for each case/assay separately.
For statistical significant results the type of analysis is provided (1-way ANOVA). p-values are always identical for 1-way ANOVA, Scheffé, or Fisher post hoc. t-test and repeated measures were calculated but not explicitely described in the figures or results.
Round 2
Reviewer 1 Report
Comments and Suggestions for Authors
1. The interaction of PMCA with Np comes from other cells doesn't mean they interact in the testis. Real experimental data in the testes in vivo definitely are needed.
2. Reference cited by authors is outdated. Upto date, there are many methods including a commercial kit to determine the Free Testosterone.
Comments on the Quality of English Language
No Comments.
Author Response
Dear Reviewer,
we thank you very much for your comments on our manuscript, although we feel that these could be better addressed in a follow up study.
Given all the accumulated evidence from Drosophila, with only a single basigin/neuroplastin gene and a single PMCA gene, to human with distinct neuroplastin and basigin genes and 4 different PMCA genes (ATP2B1-4) the formation of PMCA-Np complexes has been amply demonstrated from muscle to neuron's by groups throughout the world.
In the abstract of our manuscript, we refer to the absence of PMCA-Np complexes in Nptn-/- mice, otherwise we only refer to PMCA-Np complexes only with reference to the literature.
Within our manuscript, we do not mention PMCA1-Np complexes at all and do not refer to PMCA1-Np complexes in Leydig cells. We exclusively describe reduced expression of PMCA1 in Leydig cells in Nptn-/- mice.
As we do not claim the existence of PMCA1-Np complexes in Leydig cells, proving these goes way beyond our study. Given the very limited amount of PMCA1 in Leydig cells, an immunoprecipitation from testis extracts is possible but not as trivial as it might appear as Np and PMCA are transmembrane proteins. Even if achieved, the co-immunoprecipitation of Np and PMCA may not be regarded as proof of complex formation in Leydig cells in vivo (the detergents required for solubilization could allow artificial vicinity from different subcellular compartments). Nevertheless, the information potentially gained by such an experiment would not substantially weaken or support our hypothesis.
With regard to free testosterone, in addition to my previous response explaining the strict correlation of free testosterone and bound testosterone determined by the binding constants of SHBG and albumin and the difficulties of reliably determining free testosterone (the ELISA must be 100 times more sensitive than the total testosterone ELISA for the same specificity), the breeding and collection of new animals over the entire age range from infancy to senescence as displayed in Fig 7d for total testosterone will easily require 6- 12 months and would delay this manuscript in our opinion inappropriately.
Furthermore, some more recent references recommend the use of more sophisticated methods than an ELISA.
"Free testosterone level should ideally be measured by equilibrium dialysis method"
DOI: 10.1016/j.ecl.2021.11.002
Accurate Measurement and Harmonized Reference Ranges for Total and Free Testosterone Levels. Jasuaja et al.
" equilibrium dialysis followed by direct assessment with a trusted method is the gold standard technique for measuring free testosterone"
DOI: 10.1016/j.ucl.2022.07.009
Testosterone Assays. King B, Natale C, Hellstrom WJG.
"Currently, however, clinical possibilities for free hormone diagnostics are limited. Direct immunoassays are inaccurate, while gold standard liquid chromatography with tandem mass spectrometry (LC-MS/MS) coupled equilibrium dialysis is not available for clinical routine. Calculation models for free testosterone, despite intrinsic limitations, provide a suitable alternative, of which the Vermeulen calculator is currently the preferred method."
Role of sex hormone-binding globulin in the free hormone hypothesis and the relevance of free testosterone in androgen physiology.
Narinx N. et al, DOI: 10.1007/s00018-022-04562-1